# An Investigation of the Experiences of Working with Multilingual International Students among Local Students and Faculty Members in Chinese Universities

**Yawen Han [1], Wenxuan Li [2,*], Min Bao [1] and Xinyu Cao [3]**

[1] School of Foreign Languages, Southeast University, Nanjing 211189, China; harryhanyawen@126.com (Y.H.); ellen@seu.edu.cn (M.B.)
[2] UCL Institute of Education, University College London, London WC1H 0AL, UK
[3] College of Foreign Studies, Nanjing Agriculture University, Nanjing 210014, China; caoxinyu@njau.edu.cn
[*] Correspondence: wli25@uclan.ac.uk

**Abstract:** In recent years, as a response to the internationalization of higher education worldwide, China has begun to enroll international students to study at the tertiary level on an increasingly large scale. While the majority of the programs and courses are open to international students via Chinese as Chinese-medium instruction (CMI), there are also an increasing number of programs and courses delivered through English-medium instruction (EMI). In order to understand higher education multilingual contexts, this qualitative study examines how local students and faculty members make sense of their engagement with international students in three Chinese universities. In the study, we conducted in-depth interviews with 11 academics who worked with international students as project supervisors and 25 Chinese university students regarding their experiences of working with international students. The findings that emerged from the thematic analysis revealed that international students' learning engagement was profoundly mediated by language barriers, cultural assumptions and the academic conventions in host institutions. The study revealed that Chinese academics are concerned about international students' learning attitudes, their academic progress and a lack of participation due to their language ability. Local Chinese students also reported a lack of satisfaction in working with international students. Some of the local students felt that some international students may have been enabled to enroll in the academic programs as a result of national and university policies, which has led to a 'dumbing down' of the curriculum offered in English. The findings indicate that more needs to be done to promote mutual exchanges and better understanding among international students, Chinese faculty members and local students.

**Keywords:** multilingualism; internationalization; international students; Chinese higher education

## 1. Introduction

The globalization of higher education has triggered a large number of students undertaking academic studies abroad. This growth in the number of international students has driven a proliferation of studies exploring their learning and sociocultural experiences, including language learning [1], sociocultural adaptation [2,3], motivation for academic studies [4] and academic or living satisfaction [5]. These have been largely related to differences in cultures, systems, pedagogical and communication styles between their home and host contexts.

However, most existing studies on international students have been conducted in English-speaking countries, with only few focusing on issues related to international students in non-Anglophone

countries such as China [6–9]. Over the past decade, China has emerged as one of the most important source countries for international students. China also aspires to be a country that receives a large number of international students [10]. Since the launch of the 'Belt and Road' initiative [11], the Chinese government has heavily invested in the recruitment of students from the 'Belt and Road' countries in Central Asia, Central and Eastern Europe and Africa. In 2018, the total number of international students studying in China was nearly 500,000, about 10 times the number in 2000. Another notable rise has been the number of degree-seeking students in China [12], in particular at post-graduate level. Figures show that in 2018, 25,618 full-time international doctoral students and 59,444 Master's students studied in China—figures which used to be minimal in Chinese higher education institutions. These particular international students engage with local students and faculty members, and the increase in their numbers underscores the need to explore how they work with local students and faculty members during their terms of study. In short, the large influx of international students, particularly a sharp increase in the number of degree-seeking international students, has increased the linguistic, cultural and academic diversity in Chinese higher education institutions.

This study was implemented against the backdrop of the recent rise in the numbers of international students in China, which has raised growing concerns about the sustainable development of their education. Chinese universities have a long tradition of attracting international students to study the Chinese language, but over the last two decades they have also been promoting the use of English as the medium of instruction, in order to attract more international students as a major component of their internationalization efforts. While the majority of programs and courses are open to international students via CMI, there has been a significant increase in the number of programs and courses delivered through EMI, as well as students who choose EMI as a learning route [13].

While the expansion of the international student population creates opportunities for Chinese education institutions, it also creates a series of challenges. For example, since China is not traditionally a setting in which English is commonly used and the Chinese language is not an academic lingua franca, controversies over the learning outcomes of international students in China have been frequently raised, leading to varying perceptions and practices towards international students. These controversies pose a threat to the sustainable development of internationalization strategies in Chinese higher education.

In addition, the majority of studies on this topic tend to highlight the impact of cultural differences, asserting that international students must acquire a range of skills, awareness and attitudes—in other words, competences—in order to become fully intercultural and communicate effectively in the host context. Besides, most of these studies have been conducted through the lens of international students. As such, these has been less insight into the experiences of local students and university faculty members who are teaching and learning alongside international students. The current literature suggests that peer learning and teaching supervision has been recognized as an effective approach to supporting international students in their academic, social and emotional adjustment while they are pursuing higher education in a foreign context [14–16]. International students have also reported bullying and racial discrimination, or unequal power relationships when working with local students and teachers [17,18].

For these reasons, this paper adopts a socio-constructivist perspective and has focused on the experiences, perceptions and viewpoints of the local students and supervisors that international students meet and learn alongside, rather than merely stressing the primary roles of the defined and distinct cultures existing in their working environment. In doing this, local students and supervisors are viewed from the perspective of interculturality, as "a fluid process of being and becoming as well as describing an existing context and situation" [19] (pp. 309).

This paper addresses the following questions:

1. What challenges do local students and faculty members experience when working with international students?
2. How well do international students respond to these challenges, according to their perceptions?

### 1.1. International Student Education at Chinese Higher Education Institutions and Its Pro-Multilingual Approach

Over the last few decades of the twentieth century, the number of international students in Chinese universities rose steadily. Ever since national reforms and opening-up policies were adopted in 1978, policies for the enrolment and education of international students have gradually become more open and active, with a shift from diplomatic aims in relation to foreign countries in Asia, Africa and Latin America towards fostering international exchanges and cooperation among higher education institutions worldwide [20]. More higher education institutions have been given freedom to enroll and cultivate international students. This has contributed to a steadily growing number of international students coming to China over the past few years.

The boost in inward international students over the past two decades, aside from professional and economic opportunities [6], has also benefited from China's concerted efforts to advance the internationalization of Chinese universities. In 2010, China launched its *Study in China* plan with the aim of enrolling 500,000 international students by 2020. Meanwhile, the Chinese government provides a variety of scholarship opportunities for international students, including Chinese Government Scholarships, Local Government Scholarships, Confucius Institute Scholarships and University Scholarships. The Ministry of Education reportedly spent 3.38 billion yuan (US$481 million) in 2018 on its programs for international students in China, an increase of 16.8% over the previous year [21]. This huge investment in scholarships has had a direct impact on attracting a rising number of international students from worldwide, but in particular from South and Central Asia, Central and Eastern Europe and Africa, which account for nearly 75% of the whole population of international students at Chinese institutions.

Behind this explosive growth in international student numbers at Chinese universities lies China's pro-multilingual approach to international student education, which was reflected in the government's renewed investment in developing more programs and courses for international students, primarily delivered in Chinese and English. On the one hand China has vigorously promoted provisions for the education of international students in Chinese. By the end of 2018, Chinese universities were offering more than 40,000 CMI programs for international students [22]. On the other hand, in the context of English as a global lingua franca, the Chinese MOE also began to advocate EMI teaching [23]. Since 2010, when the MOE announced its ten-year plan for expanding international student education, there have been a series of supporting schemes such as the launch of EMI and bilingual programs (in most cases English and Chinese) for international students in many universities. By the end of 2018, 7000 EMI programs and 500 bilingual programs had been made available for international students in universities across China [22]. This rapid expansion in EMI and bilingual programs provides international students with alternatives that do not require Chinese proficiency for entrance into programs, thus increasing the potential for student recruitment.

However, an intermediate level of Chinese proficiency is in fact a necessary graduation requirement for international students taking EMI programs at Bachelor or postgraduate levels. In addition to CMI, EMI and bilingual models, there are also other foreign languages such as German, Japanese and Russian that are used as the medium of instruction for certain programs for international students at language-oriented universities [22]. This pro-multilingual approach towards the language used as the medium of delivery in courses for international students has been effective, not only in consideration of the various backgrounds and needs of international students, but also for sustaining and improving the vitality of the Chinese language in higher education communities. However, it also reflects the tension between global English and LOTE education in higher education worldwide [24].

### 1.2. Intercultural Communication and International Students' Challenges and Coping Strategies

Intercultural communication has been widely considered as a fundamental theory for studies on international students, since it ties in with the changing or changed environments in which international students live and study. In recent years, there has been a focus on an improved understanding of international students in terms of the challenges and coping strategies they encounter. Most of these studies have deployed

theories of culture shock [25], cultural adaptation [26] and intercultural competence [27]. Specifically, culture shock theories involve broadly different psychological stages and specific emotions in international students, such as anxiety, uncertainty, distress, ambiguity and frustration, all stemming from adjusting to a new environment. Cultural adaptation theories focus on how international students reorient themselves to the cultures of host countries to achieve acculturation, while intercultural competence theories stress the roles of cognitive, affective and behavioral skills in effectively and appropriately communicating with host nationals. Some recent studies [19,28] have placed a greater focus on interculturality, i.e., international students' sensitivity and understanding in cultural encounters, rather than merely on adapting to other cultures.

Extensive research has been also conducted on the challenges encountered by international students [29–31]. In addition to some common issues, such as living conditions, food differences and financial problems, the primary challenges faced by international students include language barriers, academic integration and socio-cultural adaptation [32–34].

Language barriers have been widely identified as influential in international students' academic performance and achievement, as well as in their efforts to socialize with host nationals [35]. International students' language barriers are linked to their listening and oral communication, knowledge of local contextual references, and academic writing in the target language. A lack of language proficiency can lead students to experience lower levels of security [35], self-efficacy [36] and self-esteem [37]. Language barriers can undermine international students' adjustment to the host country context [35], and might also contribute to vulnerability in either their academic or social lives [38].

Academic integration presents additional challenges to international students since they need to adjust to the host country's teaching and learning styles and must perform and be assessed according to the host country's educational values and practices. Recent studies report that difficulties surrounding international students' academic integration are related to many aspects of their performance in academic discourse, including understanding academic and grammatical jargon, meeting assignment deadlines and expectations, assessments, language proficiency, and socialization with faculty and local peer students in academic contexts [39,40]. This often stems from a lack of knowledge of the local academic environment, or a lack of confidence in using the required language in host educational contexts.

Socio-cultural adaptation is another major barrier for international students. International students are likely to encounter social support, which has a significant effect on developing relationships between international students and host nationals. Studies by Wu and Hammond [34] and Jon [18] showed that international students often experience difficulty making friends with host nationals or in communicating effectively with their teachers or local peers on campus, which is linked to cultural differences such as social interaction styles, cultural assumptions, cultural stereotyping and discrimination.

These three challenges are distinguishable, but they are also closely interdependent and interrelated. For example, the academic integration of international students involves a process of language socialization [40,41], through which they develop "the ability to communicate competently in an academic discourse community; this encompasses reading, evaluating information, as well as presenting, debating and creating knowledge through both speaking and writing" [42] so as to gain full membership in their disciplinary communities. Similarly, international students' socio-cultural adaptation is directly linked to their academic integration and language proficiency; language proficiency can facilitate, impede or regulate interactions between international students, local staff and peers. Failure to overcome these challenges may have negative effects on international students, which are linked to acculturative stress in the form of homesickness, stress, distress, anxiety, lowered self-esteem, feelings of discrimination or worsened mental health [30,43].

Many studies have discussed how international students respond to these challenges, most of which have focused on stress management strategies and cultural learning [44], discussing aspects such as adequate preparation, social participation, realistic expectations, intrinsic motivation, peer support, cultural awareness and communication skills, and avoidance [2,31,45]. All of these factors influence outcomes for international students.

However, there is less research on how these abovementioned challenges and responses are evaluated by local peers and teachers. In this paper, by exploring how local staff and peer students perceive international students, we hope to better understand the transitional process of international students in Chinese higher education.

## 2. Methodology

This exploratory study focused on local Chinese students' and faculty members' experiences of working with international students. It is hoped that this will provide an interactive and holistic perspective on the learning experiences of international students within the Chinese higher education context, and, in particular, will shed light on how Chinese teachers and students might view working with international students within their academic communities. The study involves the use of semi-structured interviews with 11 supervisors and 25 students in three different Chinese universities.

### 2.1. Participants

Purposeful convenience sampling was used to recruit participants for data collection. The participants in the study consisted of 11 Chinese MA and PhD supervisors and 25 local students across various disciplines within three Chinese universities. All the participants in this study were Chinese.

Three universities in three separate cities in China, all with considerable international student populations and long histories of recruiting international students at all levels, were selected. For the purpose of this research, the target population was supervisors and local Chinese students studying or working closely with international students. In terms of graduate-level academics, 11 faculty members who were supervising international students were invited, including two lecturers, two senior lecturers, and seven professors. Their supervision experience with international students varied from 2 years to 21 years, as shown in Table 1. In addition, 25 students, including 6 undergraduates, 8 post-graduates and 11 PhD students across a variety of academic disciplines, were also invited to participate, as shown in Table 2. Participants were not divided by university, with the aim of acknowledging teachers and students' perceptions as specific sub-entities without seeking to differentiate between universities for comparison purposes. The English language levels of the teacher participants were all self-assessed, and because all the student participants reported their English level as proficient or highly proficient, English level was not used as an indicator for the student interviews. To protect their personal information, all the people and places involved in this study have been anonymized.

**Table 1.** Profiles of teacher participants.

| Name | Gender | Department | Position | English Level | Number of International Students under Supervision at Present | Years of Supervising International Students |
|---|---|---|---|---|---|---|
| Supervisor 1 | Male | Engineering | Professor | Intermediate | 2 PhD students | 9 |
| Supervisor 2 | Female | Humanities | Professor | Intermediate | 1 PhD and 4 MA students | 21 |
| Supervisor 3 | Female | Engineering | Professor | Intermediate | 3 PhD students | 6 |
| Supervisor 4 | Male | Physics | Professor | Intermediate | 2 PhD and2 MA students | 6 |
| Supervisor 5 | Male | Environment | Lecturer | Not good | 5 MA students | 2 |
| Supervisor 6 | Male | Agriculture | Senior Lecturer | Intermediate | 4 PhD students | 5 |
| Supervisor 7 | Male | Management | Professor | Advanced | 4 PhD and 5 MA students | 11 |
| Supervisor 8 | Male | Education | Professor | Intermediate | 3 PhD and 2 MA students | 5 |
| Supervisor 9 | Male | Aerodynamics | Lecturer | Intermediate | 4 MA students | 2 |
| Supervisor 10 | Female | Economics | Professor | Intermediate | 2 PhD students | 3 |
| Supervisor 11 | Male | Agriculture | Senior Lecturer | Intermediate | 3 PhD and 1 MA student | 4 |

**Table 2.** Student participant profile.

| Number | Gender | Age | Major | Program | Number | Gender | Age | Major | Program |
|--------|--------|-----|-------|---------|--------|--------|-----|-------|---------|
| Student 1 | Female | 23 | English | MA | Student 14 | Male | 25 | Medical | PhD |
| Student 2 | Male | 31 | Electronics | PhD | Student 15 | Male | 26 | Medical | PhD |
| Student 3 | Male | 26 | Automation | PhD | Student 16 | Male | 24 | Acupuncture | MA |
| Student 4 | Female | 28 | Agriculture | PhD | Student 17 | Female | 23 | Education | MA |
| Student 5 | Female | 19 | English | BA | Student 18 | Female | 21 | French | BA |
| Student 6 | Male | 21 | Tourism | BA | Student 19 | Male | 24 | Agriculture | MA |
| Student 7 | Male | 34 | Medical | PhD | Student 20 | Male | 25 | Literature | PhD |
| Student 8 | Male | 26 | IT | PhD | Student 21 | Male | 27 | Construction | PhD |
| Student 9 | Female | 20 | Agriculture | BA | Student 22 | Male | 28 | Physics | MA |
| Student 10 | Female | 24 | Chinese | MA | Student 23 | Female | 21 | Music | BA |
| Student 11 | Male | 20 | Business | BA | Student 24 | Male | 25 | Environment | PhD |
| Student 12 | Male | 29 | Civil Engineering | PhD | Student 25 | Female | 26 | IT | MA |
| Student 13 | Female | 28 | Animation | MA | | | | | |

## 2.2. Semi-Structured Interviews

Data collection was based on in-depth semi-structured interviews with participants. Semi-structured interviews are one of the most widely-used methods of data collection within the social sciences [46] and are appropriate for studying people's perceptions and opinions or emotionally sensitive issues [47]. Taking this approach enabled the study to explore and be sensitive to the meanings that the participants attached to their own experience of international education within the broader context of the school community. Our semi-structured interviews were guided by a set of questions about the participants' experiences of and beliefs about working with international students (see Appendix A). The questions had been piloted earlier to make sure the questions were clear for participants, and the interview was carried out at an appropriate pace. The main questions focused on the following three issues: attitudes about the enrolment of international students, the difficulties of international students, and how they coped with these difficulties. Several other questions were included to gather background information. We conducted these interviews in Chinese in a one-to-one, face-to-face format. Each interview lasted around 60 to 90 min. With the participants' consent, all sessions were audio-recorded and later transcribed verbatim for analysis. In order to enhance the validity of the study, the transcripts were sent to the relevant interviewee for their comments, giving them a chance to refine what they wanted to say.

## 2.3. Data Analysis

The exploratory nature of this study necessitates inductive thematic analysis, allowing themes to emerge from the descriptions within the qualitative findings. Data analysis and interpretation followed the steps outlined in Braun [48] to utilize an inductive thematic approach. After collecting the data, we reviewed all the interview transcripts to gain an understanding of the various issues in the data. We kept an open mind to generate as many themes related to the participants' feelings and experiences of working with international students as possible. Then, informed by previous studies, we questioned and compared the relevant data to work out a list of preliminary themes. For instance, Sawir et al. [35] noted that language barriers affected international students' communication when studying abroad. This helped us to generate themes during the analysis of relevant data. When one student was asked about why Chinese students seldom interacted with international students, she gave the reason as follows:

*Well, the first reason may be, because they are not very confident about their pronunciation . . . some of them feel that their oral English is not very good, and don't know how to express themselves in English well. On the other hand, some foreign students' oral English is bad as well, especially the students from African countries. Their accents are really hard to understand. So for these Chinese students, they cannot understand what the international students say, and international students can't understand what they say too. I think this could be one reason for limited interaction between international students and Chinese students.* (Student 21)

This statement indicates that this student perceived inadequate English proficiency as a major challenge for international students. The English proficiencies of both the international students and the local students could impede their interaction. This extract also indicates that heavy accents on both sides further impinged on the communication processes.

Using the list of preliminary themes as a framework, we further coded, compared, and analyzed the data to establish specific themes and identify common patterns across them. Once such common patterns had been identified, similar themes were combined together into categories. The reiterative analysis of the data involved multiple readings of the transcripts and a progressive refinement of emerging categories. During the analytical process, we checked the consistency of coding and categorizing through regular discussions.

## 3. Findings

The data from this study showed that Chinese supervisors and local students reported a variety of complex views related to working with international students in Chinese higher education institutions. Emerging from the data analysis, the Chinese teachers and local students in the study reported that they experienced language barriers, different cultural assumptions in student-teacher interactions, and different interpretations of academic conventions as major challenges when working with international students. The findings are presented in terms of these three major themes and the participants' perceptions of international students coping with these challenges.

### 3.1. Theme 1: Language Barriers

Among the issues raised during the interviews, faculty members reported that language barriers affected their communication with international students more frequently than any other issues. Since the faculty members work with international students as project supervisors or course teachers, most of their language-related comments are to do with the use of English between themselves and the international students. Of the 11 supervisors in this study, 10 had at least one year's visiting experience in English-speaking countries, and only one had obtained his degree in Japan. However, when reflecting on their own English proficiency, most of them reported their English level as intermediate. They confessed that the use of English was challenging in their supervision process. For instance, one faculty member admitted that "academic English is a kind of a 'third language' for both teacher and student" (Supervisor 4). Another faculty member reported that English was a significant barrier for communication between himself and international students in a bilingual course that used English as the medium of instruction:

Extract 1

*At times I am faced with a situation where we (himself and international students) could not understand each other. It was just like the Chinese idiom 'a chicken talking to a duck'.* (Supervisor 5)

As can be seen from this extract, lecturing in English is demanding, and it can cause significant pressure for Chinese teachers. Similar issues have been identified by lecturers of Chinese origin in British universities, many of whom reported that they had problems communicating with students [39]. The Chinese teachers in this study also felt that the inadequate English proficiency of international students might also contribute to this communication barrier; one of them reflected on his experience:

Extract 2

> *I think I am good at daily communication and academic writing in English. It comes to be very difficult when the level of international students is not good as well. More often than not, we had been talking at cross purposes.* (Supervisor 11)

In fact, at least five Chinese teachers made negative comments about international students' English proficiency. These comments echo those that academics made about the English language proficiency of international students, particularly Chinese students, in universities in contexts such as Australia, the UK and the USA [49–51]. In specific terms, the Chinese teachers in the study noted the variations in international students' English proficiency as follows:

Extract 3

> *I've met some international students from non-English-speaking countries with excellent English communication skills. But in my university, these kinds of students are rare. The majority of international students are from non-English-speaking countries and their English level, I have to say, is not satisfactory.* (Supervisor 3)

It came as no surprise that "international students who are still struggling with listening, reading, speaking and writing skills" cannot "conduct academic discussions" (Supervisor 11) with their Chinese teachers and local students. Together with the issues that Chinese teachers themselves face in using English as the medium of instruction, international students experienced significant challenges in comprehending lectures, guidance, or tasks for participation and engagement. One Chinese teacher even suggested that the Chinese language should be used as the medium of instruction for international students to reduce the stress for Chinese teachers (Supervisor 10). This supervisor believed that proficiency in the Chinese language should be a key requirement for enrolment, although their program is advertised as being "taught in English exclusively". From the supervisors' perspective, the use of English as the medium of instruction remains a significant challenge for some international students because their lack of English proficiency prevents in-depth engagement with their academic studies and full participation in pedagogical activities.

Chinese teachers also noticed the different efforts that international students undertook to address their language challenges. Three Chinese teachers claimed that international students made progress in English proficiency as they actively sought opportunities to interact with other students and instructors. Through regular communication and social integration with fellow students, these students managed to improve their proficiency in oral English:

Extract 4

> *Effective communication requires both understanding what others say and being understood by others. These international students have to listen and speak in English when interacting with fellow students. This helps them in improving spoken language skills.* (Supervisor 2)

One teacher mentioned that international students actively asked for advice related to reading English materials. Reading articles and books in English helped them to improve vocabulary, grammar and written English. He described this as follows:

Extract 5

> *At the start of every semester, international students will ask me to assign some English articles and textbooks for them to read. There is no better teacher than a good book. By reading a textbook in English, these international students are interacting with the language. They often say that they feel different after reading a book in English.* (Supervisor 8)

However, 5 of the 11 Chinese teachers interviewed (Supervisors 3, 5, 6, 10 and 11) reported that some international students' English proficiency remained unchanged during their studies. They observed that these international students had a tendency to remain solely within the restricted social context of gatherings of their fellow nationals. Like many Chinese students in English medium universities [52], the self-segregation of these international students gave them few chances to interact with other students and might have impeded their progress in English proficiency.

In contrast to the Chinese teachers' preoccupation with the international students' English proficiency, only one student participant considered English language proficiency to be a major barrier for communication when engaging with international students in academic discussion:

Extract 6

*At first, I did have great difficulties in understanding international students from Pakistan or India. They had a very good grasp of knowledge and spoke quite fluently, but I did not understand what they said due to their accents.* (Student 11)

Moreover, although the student participants did report that English proficiency affected the international students' performance during their studies, they also felt that international students might have overused the strategy of asking for help from instructors. One local student noted:

Extract 7

*They (internationals) always approach professors with the excuse of 'poor language proficiency'. As long as the international students have problems they cannot fully understand, they would go and talk with professors, particularly before examinations. With the professors' help, they can complete their homework and pass examinations much easier than us.* (Student 10)

As can be seen in the extract, the student felt that international students might have used their lack of language proficiency as an excuse to seek and receive more help from teachers. To some extent, this student thinks that local students like him were disadvantaged in the process.

*3.2. Theme 2: Cultural Assumptions of Student-Teacher Interaction*

The second theme concerns the different cultural assumptions underlying the student-teacher interactions that Chinese teachers and local students have, in relation to the assumptions of international students. The data suggest that Chinese teachers expected international students to respect and follow the local cultural assumptions as Chinese students do. Such expectations seem to particularly focus on the Chinese cultural tradition that regulates the relations between teachers and students, but they may well be perceived by internationals as the Chinese teachers displaying power in managing their relationships and interactions with their students, both local and international. One Chinese teacher elaborated on this point:

Extract 8

*Treating a teacher like a parent and respecting his/her teaching has been a traditional virtue in Chinese culture. International students must figure out that the role of a supervisor needs to be seen as an absolute leader with authority.* (Supervisor 8)

All the teacher participants expressed their appreciation of the Chinese saying: 'He who teaches me for one day is my father for life'. It was clear that they expected international students to follow the sentiment embodied in this saying. Understandably, such expectations might not go well with international students, and therefore it might take them a long time to work out the norms that should guide their interactions with their Chinese teachers.

Interestingly, Chinese students also mentioned the existence of different assumptions about student-teacher interactions in the interviews, but these differences were not between them and the

international students, but rather between their supervisors and their universities. Student participants expressed their willingness to work and help international students, and they often played an important role in bridging the gaps between Chinese teachers and international students, as mentioned by a local Chinese Master's student:

Extract 9

> *My supervisor's English is not good. I have to spend much more time on tasks such as translating or interpreting for both sides.* (Student 19)

The statement above was echoed by four Chinese teachers in this study, who reportedly relied greatly on local students to deal with many things related to international students. Within this environment, it seems that Chinese teachers expected local students to play these mediating roles between them and international students. However, the local Chinese students did not always enjoy this responsibility. Nevertheless, it is worth noting here that the Chinese students retained an open and friendly attitude towards the international students, despite the fact that they might be asked to do more than necessary in assisting them.

As for coping strategies, the data suggest that international students made few efforts to address the noticeable differences in their assumptions about student-teacher interaction, but their presence seemed to disrupt the traditional relationship between teachers and students, which often places teachers over students in a hierarchy. In line with this traditional relationship, Chinese teachers are expected to correct students' misbehavior. However, their 'foreigner identity' afforded international students more power to avoid criticism from Chinese teachers. As a result, three Chinese teachers (all professors) mentioned that they hesitated to discipline international students, even in situations where they did not fulfil the traditional expectations of students (e.g., being hardworking):

Extract 10

> *He was very lazy and had poor concentration in class. I endured him for about half a year in silence. I can criticize Chinese students severely, but I cannot criticize international students. They are treated differently from Chinese students in all aspects.* (Supervisor 10)

In the case of graduate students, Chinese teachers usually guide them in their academic pursuits during the course of their degree studies. Chinese teachers will introduce research topics and provide all the necessary guidance and support for the research design. Deference to authority is expected of students, so that it is almost impossible for them to say 'no' even if the research design seems unrealistic. Chinese students usually comply with the decisions of the professor, but some international students might insist on their own opinions; one Chinese student reflected on the difference between him and his classmate from Africa when their views differed from those of their teacher:

Extract 11

> *Sometimes I don't agree with Prof. Chen as to his research design. However, he is an experienced supervisor and has supervised many PhD and MA students successfully. I choose to trust Prof. Chen even if I do think it is unrealistic intuitively. He is the driver, I am the passenger, and I only need to sleep in the back seat, though I don't know whether he is on the right track. I'd rather follow him than challenge him directly. But for these international students, they would refuse without hesitation if they think it is impossible. They don't like to make a compromise. Compared with them, we take a milder way to deal with the discrepancy with supervisors.* (Student 12)

As well as the research design, teachers have more control over publication, and can decide what results they choose to report. Their authority implicitly silences students' voices if they disagree. However, international students sometimes resist being silenced according to these unstated norms. One local student described how his international classmate voiced his opinion in their group meeting:

Extract 12

> *We found a result with no statistically significant difference in our experiment. Our supervisor asked us not to report it and said it was unlikely for a paper with this result to be published. To my surprise, he [the international classmate] insisted that findings with no significant differences should also be reported in our group meeting. He said it was common for researchers to find a non-significant difference, and that results should be reported. To be honest, I admire him for that.* (Student 21)

This challenge, also reported by Chinese teachers when working with international students, presents valuable opportunities for local students to reflect on their own interactions with Chinese teachers. Though such reflections might not necessarily change the ways in which local students respond to Chinese teachers, the presence of international students may subtly undermine the cultural norm of the relationship between students and teachers.

By their behavior, international students may also be likely to change local students' ways of doing things in other areas. One local student described how an international student changed local Chinese students' practices of financial reimbursement when they worked in the same project laboratory:

Extract 13

> *When we bought raw materials for our lab, he reimbursed every penny he spent on raw materials, but he never asked for more than what he actually spent. Our Chinese students would ask the dealer to round up figures and issue an invoice for 100 yuan if we spent 99 yuan. He never did that. His attitude towards money was very impressive. Because of him, we had no choice but only reimbursed for what we spent.* (Student 3)

In this situation, although the international student did not change himself in response to the challenges associated with different cultural assumptions underlying the student-teacher interactions, he did change some of the ways in which Chinese students and teachers interacted. This international student influenced the local students' perceptions and practices.

*3.3. Theme 3: Academic Socialization*

Academic socialization emerged as another recurring theme in the data analysis. In this study, this concept reflects how international students meet expectations for assessment and teaching and adopt learning pedagogies from the host academic context. The findings suggest that discrepancies between the local institutional cultures and international students' beliefs were frequently mentioned as a significant challenge for international students by both the Chinese teachers and the students. One teacher suggested that international students were too culturally disconnected to form "meaningful" relationships:

Extract 14

> *I think they [international students] should know that one of the traditional key principles in Chinese education is to learn to be an upright person and learn how to get along with others. So, there's no need to stress out about how different you are in this group.* (Supervisor 2)

Likewise, another teacher complained that international students are sometimes "too free and always follow their own inclinations." The Chinese teachers expected international students to adjust themselves to fit the group, in this case, the academic community in Chinese universities. In contrast, the majority of Chinese students did not share the teachers' expectations of international students, as they felt they were "from a much newer generation and do not strongly believe in 'collectivism'" (Student 9).

Though Chinese teachers and students may have different expectations of international students, they did share similar values associated with academic studies, such as being hardworking and punctual. Besides the teaching pedagogy, the norms for Chinese teachers and students may also present challenges

for international students in their academic socialization. For example, one local student complained that international students were ignorant of their regulations in the lab:

Extract 15

> *In our lab, this is an unwritten rule: when we do our experiment, we take turns to treat lunch. Today I treat you, and tomorrow you treat me. But in their eyes, it is a matter of principle. They refuse to follow this rule because they think it is deviating from their values.* (Student 2)

Nevertheless, some international students did take proactive measures to socialize themselves into the local academic community. The most common strategy they used was to offer help with research expertise in experiments, which helped to strengthen their partnerships with fellow students in the academic community:

Extract 16

> *Songa is good at analyzing data with the software Graphpad Prism. He is very popular in our lab. He helps us analyses our data, and we like to discuss our research with him.* (Student 24)

Although interaction with faculty members would enhance the international students' academic adaptation into the academic community, most teacher participants reported little contact with their international students after class or outside the lab. One teacher commented:

Extract 17

> *I don't spend much time with them. When I meet them, we just talk about research. I have no time to chat with them about other things, and I am not good at chatting about non-academic stuff.* (Supervisor 1)

In China, teachers often have dinner with their students, but most of the teachers in the study seldom ate out with international students. This lack of socialization between teachers and international students might have to do with language issues and differences in cultural assumptions about student-teacher interactions. Other factors such as dietary differences may also have contributed. It seems that these cultural gaps may take a long time to overcome, involving significant efforts on the part of both Chinese teachers and international students.

## 4. Discussion

This paper examines how Chinese teachers and local Chinese students reflected on their experiences of working with international students. It generates useful implications for the development of higher education in the process of multilingual internationalization in China. To the best of the researchers' knowledge, the issue of academic engagement and the perspectives of teachers and local students on international students have not been the subject of much investigation in the literature on international students. This study, thus, fills these gaps with reflections on international students from teachers' and local students' perspectives, allowing an articulation of what happens in the process of working with real students.

As mentioned earlier, the analysis helped us to identify three major issues in the participants' reflections: language barriers, cultural assumptions, and academic socialization. It is likely that these perceptions on the part of Chinese teachers and local students will mediate their approaches to working with international students, and hence, the educational experiences of international students in Chinese universities. For some local students and supervisors, how they perceived international students in their academic lives also accentuated the significance of interculturality in the process of cross-cultural communication.

Previous research has found that language barriers affect both international students and their teachers academically [39,53,54]. This study confirms the significance of language, identifying language barriers as a major impediment to mutual communication between Chinese teachers and international students. Insights from these teachers' accounts showed that insufficient English levels for some

international students, as well as their supervisors, hindered academic engagement on both sides, limiting academic exchange to lectures and workshops with no deeper and or more meaningful exchanges while socializing between Chinese teachers and students. The study noted that some international students were found to have made efforts to address this challenge. Nevertheless, it seems that Chinese teachers, local students, and international students may all have insufficient English for academic socialization in the medium of English, which begs the question of the purpose of having English as the medium of instruction in Chinese universities.

The study also identified differences in the cultural assumptions that Chinese teachers, local students, and international students have with regard to student-teacher interaction, which may also constitute a significant challenge for the participants. In particular, most of the Chinese teachers in the study worked with international students as their project or thesis supervisors. The arrival of international students challenged the hierarchical nature of student-teacher interaction that many Chinese teachers are used to [55], which might have caused Chinese teachers to experience dissatisfaction in their interactions with international students. Local students were sometimes asked by Chinese teachers to mediate their interactions with international students, and this mediation encouraged local students to reflect on their own interactions with teachers. This may have led them to become critical about the extra help international students received from teachers [56]. It may also have made them realize that alternative ways of interacting with teachers are possible, as they witnessed international students challenging their Chinese teachers.

The final theme is to do with academic socialization, as international students may face significant challenges adjusting to these local norms. This finding corroborates previous studies showing that international students are likely to encounter a number of academic issues as they live and study in a different culture [57–59], among which teaching pedagogy is significant. However, this study also highlighted that apart from teaching pedagogy, there are many other conventions, both overt and implicit, which international students struggle to deal with. These conventions are likely to be familiar to local Chinese students, but international students may think them strange or feel uncomfortable with them. The study also noted that international students made efforts to socialize with local students by investing their skills in mutually beneficial and sustainable exchanges.

## 5. Implications and Limitations

The findings in this study highlight several key factors pertaining to Chinese teachers' and local students' perceptions of international students that are prominent in, but may not be unique to, the multilingual context of Chinese higher education. However, these findings need to be treated with caution as they might be different if this study were to be repeated in other Chinese universities, where the conditions for Chinese teachers, local students, and international students might be very different compared with those in the universities in this study.

Nevertheless, the findings have several implications for policymakers in international education in Chinese higher education, as well as a number of practical implications to help Chinese teachers and students communicate more effectively with international students. First, it seems necessary to make efforts to bring international and local students together in Chinese universities. Chinese universities tend to adopt a 'separation approach' to international students' accommodation and services, providing international students with better living conditions than local students. These policies and approaches, which help to create an 'enjoyable' environment for the international students, are designed to attract more international students to China; however, they also contribute to the seclusion of international students from local students. We suggest that policymakers, universities and supervisors should take the initiative to create an encouraging and engaging educational atmosphere, providing training courses on communication strategies to benefit all parties. In addition, a specific office is also recommended for individual Chinese higher education institutions to foster the academic adaptation of international students within their educational programs, making China's international education more substantiable.

It is argued that the mutual adaptation of international students, local students and teachers will be important for the sustainable development of international education in China.

Second, it is important for Chinese teachers and local students to re-educate themselves in light of the increasing numbers of international students. The distinct lack of cross-cultural training and English language abilities among the teachers in this study clearly impeded mutual academic engagement between Chinese teachers and international students. Chinese teachers and students in the host context need to realize that the academic acculturation of international students should be a two-way process [26], which means they also need to adjust themselves to the academic acculturation of international students and to the increasing internationalization of Chinese higher education. We also suggest that future studies should pay greater attention to how teachers and local students view international students' intercultural experiences, and how these understandings reflect their own assumptions, perceptions and practices.

## 6. Conclusions

In this paper, the perspectives of university faculty members and local students in relation to international students were examined. However, it is important to acknowledge that these findings may not accord with international students' own perceptions of their academic engagement in China. The current study is part of a larger mixed-methods research project on international students in Chinese higher education institutions, and the findings from this study will thus be triangulated with other findings from the wider research project, including perceptions about academic engagement from international students themselves, thereby providing a holistic insight into the emergent dynamics of the situation. However, it is recognized that the faculty members in this study were all Chinese, although supervisors from international backgrounds were not purposefully excluded. Given the increasing level of internationalization in Chinese higher education institutions, future studies in this field incorporating supervisors from international backgrounds would be more powerful in terms of providing in-depth and comprehensive accounts of higher education faculties as a whole.

In investigating the perceptions of Chinese staff and students and how they extracted meaning from their experiences, this study identified tensions between Chinese faculty members, local Chinese students, and international students. The findings suggest there exists a weak academic cohort of international students who cannot conduct meaningful academic exchanges with Chinese students and teachers. More efforts are thus required to further integrate these weaker international students into the context of Chinese higher education. More importantly, Chinese supervisors and students must also recognize the need to adjust themselves to the increasing numbers of international students, in order to create a truly inclusive and engaging environment.

**Author Contributions:** W.L. designed the empirical study and wrote the initial draft; Y.H. proposed the main idea and revised the paper. M.B. and X.C. collected the data of the paper; All authors have read and agreed to the published version of the manuscript.

**Funding:** This research was funded by National Office for Philosophy and Social Sciences of China (grant number 19BYY037).

**Conflicts of Interest:** The authors declare no conflict of interest.

## Appendix A

### Interview Questions for Teachers:

1. Describe your educational background. Have you ever studied or worked in another country before? How is your English?
2. Are there international students in your program? Why do you think international students choose to study in China? Why do you think the Chinese government and universities recruit international students?

3. When did you begin to supervise international students? How many international students are you supervising at present? For what reason did you decide to supervise them? What contributions can they bring to your program?

4. Is there any problem in your supervision of international students? What frustrates you the most in supervising international students? How do you solve these problems?

5. Do international students have problems during their study? What are the causes of their problems? How do they tackle these problems?

**Interview questions for students:**

1. Do you have international students in your class or lab? Why do you think these international students choose to study in China? Why do you think the Chinese government and universities recruit international students?

2. Do you often interact with international students? Why, or why not? For what reason do you decide to approach them?

3. What do you think of international students' interactions with teachers? Is there any problem? What are the causes of these problems?

4. How are international students different from Chinese students in dealing with teachers?

5. What do you think of international students' engagement in their study? What problems do the international students have? How do the international students tackle these problems?

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
