# Peer review of "An Investigation of the Experiences of Working with Multilingual International Students among Local Students and Faculty Members in Chinese Universities"

_sustainability, doi:10.3390/su12166419_

Round 1

Reviewer 1 Report

Very interesting EMI study from a different perspective. 

Very interesting article form a different point of view.

Line 50. 'Belt and road' initiative for recruiting students could be a bit more explained. Does it also affect teachers recruitment?

Line 82. Research questions 1, 2 refer only to students´ point of view whilst in the title and abstract teachers are also mentioned as part of the research.

In line 181 teachers are again mentioned as part of the study. 

Line 186, mentioning of semi-guided interviews with 11 supervisors, is it meant to be administrative staff or do they mean teachers. (specified in line 197). 

Line 191. Is gender item relevant? Purpose?

line 256. Teachers report having language difficulties when dealing with international students, does the institution require any language proficiency level to supervise international students? Teach EMI ? 

Line 264. Lingua franca could be mentioned as the "third language", English. 

Author Response

Dear Reviewer:

Thank you for your letter and for the reviewers’ comments concerning our manuscript entitled “S Promoting Sustainable Development for International Student Education in a Multilingual Space: An Investigation of the Experiences of Working with International Students among Local Students and Faculty Members in Chinese Universities”. Your comments are all valuable and very helpful for revising and improving our paper, as well as the important guiding significance to our research. We have studied comments carefully and have made correction which we hope meet with approval. We believe that the revisions prompted by these comments have strengthened our manuscript.

Revised portion are marked in red in the paper. The main corrections in the paper and the responds to reviewer’s comments are as flowing:

Point 1: Line 50. 'Belt and road' initiative for recruiting students could be a bit more explained. Does it also affect teachers recruitment?

Response: Thank you for your comments. This has now been corrected, as suggested. This initiative was closely related to the explosive growth of international students in China but has less connection to our recruitment of teacher participants in this study.

Point 2: In line 181 teachers are again mentioned as part of the study.

Response: Thank you for your comments. Teachers’ perceptions were of significance in this study. This has now been amended in the revised manuscript.

Point 3: Line 186, mentioning of semi-guided interviews with 11 supervisors, is it meant to be administrative staff or do they mean teachers. (specified in line 197).

Response: Thank you for your comments. Our study aims to explore the perceptions of local students and faculty members, with a special focus on the learning and teaching process. We do acknowledge the prominent role of administrative staff in international students’ adaptation to a host culture. However, the role of administrative staff is primarily managerial, which is not the focus of this study. We hope to explore this topic in our next paper.

Point 4: Line 191. Is gender item relevant? Purpose?

Response: Thank you for your comments. Our participants consisted of 11 supervisors, of whom three were female and eight were male, and 25 international students, of whom 10 were female and 15 were male. The gender proportion of local student participants in our study is similar to that of the populations of the three universities in this study. However, we did not purposefully balance the gender difference among supervisor participants in our sample. We adopted purposeful sampling and selected supervisors who worked closely with international students. Our focus is on teachers and students’ perceptions, regardless of their age and gender.

Point 5:Line 256. Teachers report having language difficulties when dealing with international students, does the institution require any language proficiency level to supervise international students? Teach EMI ?

Response: Thank you for your comments. Our findings indicate that the language proficiency of supervisors who delivered EMI courses or tasks at the institutions involved in this study was low. This has been amended in the revised manuscript.

Point 6:Line 264. Lingua franca could be mentioned as the "third language", English.

Response: Thank you for your comments. We have changed “lingua franca” to “third language”. This has now been amended in the revised manuscript.

Reviewer 2 Report

Research into internationalisation of higher education is essential in today's world, while the authors' decision to study local staff and students response to internationalization touches a somewhat neglected aspect of social comfort and effectiveness of the educational process. The findings indicate that the language factor cannot be overestimated in the success of internationalisation, but tends not to be given sufficient attention by administrators of HE.

The problem of low level of social mixing of international and local students is fairly universal, observed at universities around the world. Studies of the problem contribute to better understanding and finding solutions to cope with it.

Author Response

Dear Reviewer:

Thank you for your letter and for the reviewers’ comments concerning our manuscript entitled “S Promoting Sustainable Development for International Student Education in a Multilingual Space: An Investigation of the Experiences of Working with International Students among Local Students and Faculty Members in Chinese Universities”. Your comments are all valuable and very helpful for revising and improving our paper, as well as the important guiding significance to our research. We have studied comments carefully and have made correction which we hope meet with approval. We believe that the revisions prompted by these comments have strengthened our manuscript.

Revised portion are marked in red in the paper. The main corrections in the paper and the responds to reviewer’s comments are as flowing:

Point 1: Research into internationalisation of higher education is essential in today's world, while the authors' decision to study local staff and students response to internationalization touches a somewhat neglected aspect of social comfort and effectiveness of the educational process. The findings indicate that the language factor cannot be overestimated in the success of internationalisation but tends not to be given sufficient attention by administrators of HE.

Response: Thank you for your comments. Our study aims to explore the perceptions of local students and faculty members, with a special focus on the learning and teaching process. We do acknowledge the prominent role of administrative staff in international students’ adaptation to a host culture. However, the role of administrative staff is primarily managerial, which is not the focus of this study. We hope to explore this topic in our next paper.

Point 2: The problem of low level of social mixing of international and local students is fairly universal, observed at universities around the world. Studies of the problem contribute to better understanding and finding solutions to cope with it and finding solutions to cope with it.

Response: Thank you for your comments. We have added new implications in the revised manuscript.  

Reviewer 3 Report

Methodology
The study indicates that they are reflections, but these reflections are sometimes left without clear evidence, and without validating the semi-structured interviews that they say serve to arrive at those reflections
there is hardly any revision and theoretical foundation of the semi-structured interviews, nor a validation as a data collection technique
And above all: it is indicated to see in appendix A, specifically "Our semi-structured interviews were guided by a set of questions about the participants' experiences and beliefs about working with international students (see Appendix A)". However, no annex appears; an annex that becomes necessary, and that it could even be that it could go in the text itself.

Without validating the only instrument of analysis, the rest of the sections: results, discussion and analysis, lose their argumentation, validity and scope. Everything is open without that appendix A:
"The data collection was based on in-depth semi-structured interviews with participants."
Which? It remains to delimit the sample, the script of the interviews (appendix A); which went to lumnos and which to teachers
Interviews are one of the most widely used data collection methods in the social sciences
yes, but most of the time they are the complement, the interviews serve as a complement or contrast to other instruments

Results
the language variable is interesting; although sometimes this aspect is the most superficial object of study to delimit

The study also identified differences in cultural assumptions in social interaction that mediate the teacher-student relationship that Chinese teachers and international students have with respect to student-teacher interaction, which may also constitute a major challenge for the author / the authors demonstrate with evidence the study of these interactions.

The real object of study seems to be academic socialization. Perhaps the revision of updated previous studies should be increased.

Regardless of the specific questions, the interviews always addressed these issues: attitudes about the international enrollment of the 19 students, the difficulties of international students and how they faced these difficulties. Each interview lasted around 60 to 90 minutes and was conducted in Chinese. With the consent of the participants, all sessions were audio-recorded and transcribed for analysis.

Conclusions

As we do not know the script of the semi-structured interview, we cannot contrast or replicate where we start for these reflections
and, therefore, they are questioned because results and discussion are based on reflections for which we have no evidence, nor the procedure of how it has been carried out.

Author Response

Dear Reviewer:

Thank you for your letter and for the reviewers’ comments concerning our manuscript entitled “S Promoting Sustainable Development for International Student Education in a Multilingual Space: An Investigation of the Experiences of Working with International Students among Local Students and Faculty Members in Chinese Universities”. Your comments are all valuable and very helpful for revising and improving our paper, as well as the important guiding significance to our research. We have studied comments carefully and have made correction which we hope meet with approval. We believe that the revisions prompted by these comments have strengthened our manuscript.

Revised portion are marked in red in the paper. The main corrections in the paper and the responds to reviewer’s comments are as flowing:

Point 1:  Methodology.The study indicates that they are reflections, but these reflections are sometimes left without clear evidence, and without validating the semi-structured interviews that they say serve to arrive at those reflections; there is hardly any revision and theoretical foundation of the semi-structured interviews, nor a validation as a data collection technique. And above all: it is indicated to see in appendix A, specifically "Our semi-structured interviews were guided by a set of questions about the participants' experiences and beliefs about working with international students (see Appendix A)". However, no annex appears; an annex that becomes necessary, and that it could even be that it could go in the text itself. Without validating the only instrument of analysis, the rest of the sections: results, discussion and analysis, lose their argumentation, validity and scope. Everything is open without that appendix A:

Response 1:  Thank you very much for your valuable comments and feedback regarding our research methodology. The appendix has been attached to this manuscript.

Point 2: Methodology-"The data collection was based on in-depth semi-structured interviews with participants."Which? It remains to delimit the sample, the script of the interviews (appendix A); which went to lumnos and which to teachers; Interviews are one of the most widely used data collection methods in the social sciences; yes, but most of the time they are the complement, the interviews serve as a complement or contrast to other instruments.

Response 2:  Thank you very much for your valuable comments and feedback regarding our research methodology. We firmly believe semi-structed interviews with participants is the most suitable approach for this study. This approach is appropriate for studying people’s perceptions and opinions or emotionally sensitive issues (Barriball & While 1994). Also, our study is an entirely qualitative one. Being interpretive, our approach can identify possible relationships, causes, effects and dynamic processes, rather than generalize. The feature also fits well with the exploratory nature of this study; that is, to find out what was actually happening to these international students in China. We argue that interviews were conducted not only because they are a rich source of data, but also because they provided us with channels of communication with the participants so as to have ‘meanings’ negotiated interactively.

Point 3:  Results-The language variable is interesting; although sometimes this aspect is the most superficial object of study to delimit. The study also identified differences in cultural assumptions in social interaction that mediate the teacher-student relationship that Chinese teachers and international students have with respect to student-teacher interaction, which may also constitute a major challenge for the author / the authors demonstrate with evidence the study of these interactions. The real object of study seems to be academic socialization. Perhaps the revision of updated previous studies should be increased. Regardless of the specific questions, the interviews always addressed these issues: attitudes about the international enrollment of the 19 students, the difficulties of international students and how they faced these difficulties. Each interview lasted around 60 to 90 minutes and was conducted in Chinese. With the consent of the participants, all sessions were audio-recorded and transcribed for analysis.

Response 3: Thank you for your comments.

The text has now been amended to provide better clarification. We have also added “academic socialization” into our literature review and discussion.

Point 4: Conclusions. As we do not know the script of the semi-structured interview, we cannot contrast or replicate where we start for these reflections and, therefore, they are questioned because results and discussion are based on reflections for which we have no evidence, nor the procedure of how it has been carried out.

Response 4: 

Thank you for your comments. The appendix has been attached to this manuscript.

Reviewer 4 Report

The topic and the findings are interesting, however, I feel there are several issues in the manuscript that have to be addressed.

Abstract

Abbreviations must be explained, or preferably, not used, and use them later in the text. There are also some typos.

Introduction

You totally omitted the topic of interculturality (or cross-culturality), I feel it needs a very deep evaluation in your article as it deals with the same topic and your paper needs this theoretical background. There is a vast literature on the topic. You only mention one book on the topic and it is very obsolete.

What are the potential issues connected to interculturality with respect to the Chinese vs Western encounters? They will be different when compared with Chinese vs other Asian encounters. What are the consequences of cultural discrepancies? How much is interculturality important in the global academic world? Could you mention the topics of culture shock, cultural adaptation, interculturality in mode depth, please? Could you reveal the topics of interculturality in your results as well, please? I feel that it is not possible to ignore it in the paper and just focus on language barriers.

Methodology

Can you better describe your research sample? The regions?

I would add the questions of the interview into the text – I need to get some idea to show how the interviews were conducted and what they focused on.

Discussion

Again, you only mention the topic of language barriers even if the findings clearly indicated that interculturality is a grave issue. I would appreciate it if you could bring more relevant literature and research here that shows the topic of interculturality in the context of your research. Stating that language barriers affect both the teachers and the students is too obvious and maybe you could bring some more non-intuitive ideas and concepts – Sustainability is an impact journal.

Implications

I would like to see very clear recommendations and a suggestion of a trajectory of the communication strategy in the given context if this should be an impact paper. Mere theoretical recommendations are not sufficient. To be an impact paper, you need to clearly show the novelty of your findings and try to implement them into a clear policy/strategy that could be utilized by the policymakers and university authorities.

References

The references contain a lot of typos and mistakes and should be made more consistently (dash vs hyphen, sometimes pp sometimes not, bold vs not bold for years).

Author Response

Dear Reviewer:

Thank you for your letter and for the reviewers’ comments concerning our manuscript entitled “Promoting Sustainable Development for International Student Education in a Multilingual Space: An Investigation of the Experiences of Working with International Students among Local Students and Faculty Members in Chinese Universities”. Your comments are all valuable and very helpful for revising and improving our paper, as well as the important guiding significance to our research. We have studied comments carefully and have made correction which we hope meet with approval. We believe that the revisions prompted by these comments have strengthened our manuscript.

Revised portion are marked in red in the paper. The main corrections in the paper and the responds to reviewer’s comments are as flowing:

Point 1: Abstract-Abbreviations must be explained, or preferably, not used, and use them later in the text. There are also some typos.

Response: Thank you for your comments. This has now been amended in the revised manuscript.

Point 2: Introduction-You totally omitted the topic of interculturality (or cross-culturality), I feel it needs a very deep evaluation in your article as it deals with the same topic and your paper needs this theoretical background. There is a vast literature on the topic. You only mention one book on the topic and it is very obsolete.

Response: Thank you for your comments. This has now been amended in our literature review and discussion.

Point 3: What are the potential issues connected to interculturality with respect to the Chinese vs Western encounters? They will be different when compared with Chinese vs other Asian encounters. What are the consequences of cultural discrepancies? How much is interculturality important in the global academic world? Could you mention the topics of culture shock, cultural adaptation, interculturality in mode depth, please? Could you reveal the topics of interculturality in your results as well, please? I feel that it is not possible to ignore it in the paper and just focus on language barriers.

Response: Thank you for your comments. Our second theme – cultural assumptions – concerns the issue of interculturality. Regarding Chinese vs. Western encounters, instead of holding that cultural discrepancies are the most influential factor in international students’ intercultural adaptation, our paper searched for factors beyond cultural difference, which can easily lead to cultural stereotyping and bias. Therefore, we did not provide a comparison between Chinese and Western cultures or Chinese and other Asian cultures in our literature review. However, we have highlighted the role of interculturality in this paper, as suggested, and have added the topics of culture shock, cultural adaptation and interculturality to our literature review and discussion sections.

Point 4: Methodology

Can you better describe your research sample? The regions? I would add the questions of the interview into the text – I need to get some idea to show how the interviews were conducted and what they focused on.

Response: Thank you for your comments. Our semi-structured interviews were guided by a set of questions about the participants’ experiences of and beliefs about working with international students (please see Appendix).  

Point 5: Discussion-Again, you only mention the topic of language barriers even if the findings clearly indicated that interculturality is a grave issue. I would appreciate it if you could bring more relevant literature and research here that shows the topic of interculturality in the context of your research. Stating that language barriers affect both the teachers and the students is too obvious and maybe you could bring some more non-intuitive ideas and concepts – Sustainability is an impact journal.

Response: Thank you for your comments. We have highlighted interculturality in this paper. This has now been amended in our literature review and discussion.

Point 6: Implications-I would like to see very clear recommendations and a suggestion of a trajectory of the communication strategy in the given context if this should be an impact paper. Mere theoretical recommendations are not sufficient. To be an impact paper, you need to clearly show the novelty of your findings and try to implement them into a clear policy/strategy that could be utilized by the policymakers and university authorities.

Response: Thank you for your comments. We have added new implications. 

Point 7: References-The references contain a lot of typos and mistakes and should be made more consistently (dash vs hyphen, sometimes pp sometimes not, bold vs not bold for years).

Response: Thank you for your comments. This has now been corrected as suggested. 

Round 2

Reviewer 4 Report

Not all abbreviations have been explained in the abstract as suggested earlier.

The paper deals with the topic of interculturality and therefore it should be introduced in the Introduction part not as late as discussion. It is crucial to state the recent theories and practices of intercultural communication so that the paper has a solid foundation. I do not think that mentioning the topic of interculturality in the final part of the manuscript is sufficient. 

I still lack recommendations and clearer strategy in line with sustainability of university education.

The majority of new references added contain a lot of mistakes and are not consistent with the template of the paper. 

Author Response

The authors would like to thank the Reviewer for his/her valuable comments.

  1. Not all abbreviations have been explained in the abstract as suggested earlier.

Response: Thank you for your comments. This has now been amended in the revised manuscript.

  1. The paper deals with the topic of interculturality and therefore it should be introduced in the Introduction part not as late as discussion.

Response: Thank you for your comments. This has now been added to our introduction in the revised manuscript.

  1. It is crucial to state the recent theories and practices of intercultural communication so that the paper has a solid foundation. I do not think that mentioning the topic of interculturality in the final part of the manuscript is sufficient. 

Response: Thank you for your comments. We have highlighted intercultural communication in the literature review and have relocated our discussion of relevant theories. This has been very useful for interpreting the perceptions and behaviors of participants in this study.  

  1. It still lacks recommendations and clearer strategy in line with sustainability of university education.

Response: Thank you for your comments. This has now been added to the implication in the revised manuscript.

  1. The majority of new references added contain a lot of mistakes and are not consistent with the template of the paper. 

Response: Thank you for your comments. This has now been amended in our revised reference.

Round 3

Reviewer 4 Report

The authors have implemented all my suggestions and I feel the paper has been improved significantly after all the reviewers´ suggestions. 

Line 46: the abbreviation HE is not explained, again. Btw, in this sentence it does not make much sense, should be "between HE students". I guess there is no reason to use this abbreviation as it appears only a few times and only creates strange grammar structures, see further in the text. "China HEs" is what? Education is not plural, should be Chinese higher education. Can you fix these sentences, please?

There is no reason not to warrant an agreement regarding the publication of this paper. The paper is well structured and contains interesting findings. 

However, there are still issues with the references (e.g. a hyphen is not the same thing as a dash, some of the entries are followed by a full stop but some are not) but these issues could be fixed in the final stage of the manuscript editing. 

I also suggest proofreading of the manuscript as there are several minor issues. 

The last suggestion: make the title shorter, this is way too long.

Author Response

We would like to thank the reviewer for his/her time spent on reviewing our manuscript several times and the wonderful comments helping us improving the article. The suggestions offered by the reviewer have been immensely helpful.

  1. Line 46: the abbreviation HE is not explained, again. Btw, in this sentence it does not make much sense, should be "between HE students". I guess there is no reason to use this abbreviation as it appears only a few times and only creates strange grammar structures, see further in the text. "China HEs" is what? Education is not plural, should be Chinese higher education. Can you fix these sentences, please?

Response: Thank you for your comments. This has now been corrected, as suggested.

  1. There are still issues with the references (e.g. a hyphen is not the same thing as a dash, some of the entries are followed by a full stop but some are not) but these issues could be fixed in the final stage of the manuscript editing.

Response: Thank you for your comments. This has now been amended in the revised manuscript.

  1. I also suggest proofreading of the manuscript as there are several minor issues.

Response: Thank you for your suggestions. This manuscript has been proofread by a professional in this field.

  1. The last suggestion: make the title shorter, this is way too long.

Response: Thank you for your comments. The title has now been changed to “An Investigation of the Experiences of Working with Multilingual International Students among Local Students and Faculty Members in Chinese Universities”.